# Novel Corrosion Inhibitor for Carbon Steel in Acidic Solutions Based on α-Aminophosphonate (Chemical, Electrochemical, and Quantum Studies)

**DOI:** 10.3390/molecules28134962

**Published:** 2023-06-24

**Authors:** M. A. Deyab, Marwa M. Abdeen, Mohamed Hussien, Ibrahim E. El-Sayed, Ahmed Galhoum, Omnia A. A. El-Shamy, Marwa Abd Elfattah

**Affiliations:** 1Egyptian Petroleum Research Institute, Nasr City, Cairo 11727, Egypt; hamadadeiab@yahoo.com (M.A.D.); omniaelshamy@yahoo.com (O.A.A.E.-S.); 2Basic Science Department, Higher Institute of Engineering and Technology, Menoufia, Egypt; marwamaherabdeen94@gmail.com; 3Department of Chemistry, Faculty of Science, King Khalid University, P.O. Box 9004, Abha 61413, Saudi Arabia; mhalmosylhy@kku.edu.sa; 4Chemistry Department, Faculty of Science, Menoufia University, Shebin El-Kom 32519, Egypt; 5Nuclear Materials Authority, El-Maadi, P.O. Box 530, Cairo 11727, Egypt; galhoum_nma@yahoo.com; 6Chemical Engineering Department, Higher Institute of Engineering and Technology, Menoufia, Egypt; mzmi2005@gmail.com

**Keywords:** organo-inhibitors, acid solutions, steel, corrosion, adsorption

## Abstract

α-aminophosphonate (α-AP) is used as a novel corrosion inhibitor for carbon steel. The aggressive media applied in this study are HCl and H_2_SO_4_ acid solutions. The findings indicate that the morphology of the α-AP compound is cubic, with particles ranging in size from 17 to 23 μm. FT-IR, ^1^HNMR, ^31^PNMR, and ^13^CNMR analysis confirmed the synthesis of the α-AP molecule. It has been discovered that the compound α-AP plays an important role in inhibiting the corrosion of carbon steel in both HCl and H_2_SO_4_ acids. This was identifiably inferred from the fact that the addition of α-AP compound decreased the corrosion rate. It is important to report that the maximum inhibition efficiency (92.4% for HCl and 95.7% for H_2_SO_4_) was obtained at 180 ppm. The primary factor affecting the rate at which steel specimens corrode in acidic electrolytes is the tendency of α-AP compounds to adsorb on the surface of steel through their heteroatoms (O, N, and P). This was verified by SEM/EDX results. The adsorption actually occurs through physical and chemical mechanisms via different active centers which are matched with the calculated quantum parameters. In addition, the adsorption of α-AP follows the Langmuir isotherm.

## 1. Introduction

The main engineering material used to construct pipelines for transporting water, chemicals, and petroleum products, and the vessels used in oil and gas production systems is carbon steel (C-steel). Due to its affordable price and superior mechanical qualities, carbon steel is widely used [1]. Regrettably, objects made of carbon steel are susceptible to corrosion by the atmosphere around them [2]. Among the most urgent issues facing the gas and oil sector is corrosion, which causes the mechanical qualities of the apparatus to deteriorate leading to leakage of oil into the environment [3,4]. The steel pipe wall suffers severe corrosion during acid washing, causing material corrosion and a reduction in pipe strength [5]. There is a trend toward replacing alloy equipment with more corrosion-resistant materials, but unfortunately, this raises production prices. New corrosion inhibitors that prevent material deterioration by depositing a resistive coating on the surface of the metal are one of the primary answers to this issue [6,7,8]. Generally, inhibitors are mixed with the acids during the cleaning process as a first line of defense [9]. The group of different corrosion inhibitors, which includes inorganic, organic, and heterocyclic compounds, is the most significant source that is used to protect steel pipes [10].

Organic compounds, particularly those with a heteroatom and/or π electrons in their composition, have drawn much attention as metal corrosion inhibitors because they can easily adsorb and form a resistive layer on the metal surface that stops the aggressive agents in the environments to penetrate and attack its surface [11,12]. Additionally, they are less harmful to humans and the environment than inorganic inhibitors [13].

It is common knowledge that a suitable corrosion inhibitor must function well even at low doses. Phosphonates are valuable and intriguing inhibitors due to their stability toward hydrolysis, low toxicity, activity in aqueous conditions, and resistance to microbial degradation [14] because of their sp3 hybridized nitrogen atoms which increase the inhibition effect [15]. The following are widely utilized as corrosion inhibitors: phosphorus compounds such as phosphonic acids [16], aminomethylene phosphonates [17], pyrazine derivatives [18], α-aminophosphonates [19], and nitrogen-containing molecules such as imidazolines, amides, and amidoamines [20]. Due to their outstanding coordination properties with metal ions, organophosphorus compounds, and derivatives such as α-aminophosphonates (contain P, N, and O as heteroatoms), have been thoroughly researched for metal recovery, water purification, and metal extraction [21,22,23]. For α-aminophosphonates, numerous synthesis methods have been reported. The three-component reaction of carbonyl compounds, amines, and phosphite moiety in the presence of a Lewis acid catalyst is the most efficient, straightforward, and high-yielding method [24]. The study of reaction mechanisms has frequently utilized quantum-chemical calculations. It has also been demonstrated to be a valuable tool for researching corrosion control mechanisms [25]. The introduction of density functional theory (DFT) has significantly improved the use of quantum calculation as a supportive tool in developing organic corrosion inhibitors. Although different compounds have been tested in the laboratory or by incrementally changing the structures of existing inhibitors, corrosion scientists have historically found new corrosion inhibitor molecules using these methods, which are frequently costly and time-consuming [26].

The corrosion inhibition in carbon steel was investigated in sulfuric and hydrochloric acid solutions; incorporating α-aminophosphonate (α-AP) as a corrosion inhibitor is the subject of the work discussed in this paper. The inhibitors’ effectiveness is determined at varied doses. Furthermore, theoretical calculations using DFT are applied to illustrate the relation between the electronic structure and corrosion behavior of α-AP.

## 2. Results

### 2.1. Characterization of the Synthesized α-AP Compound 

#### 2.1.1. SEM Analysis

Figure 1a shows the morphology observation for the α-AP compound, and Figure 1b shows the cubic shape with particle size range from 17–33 μm.

#### 2.1.2. ^1^H NMR, ^31^P NMR, and ^13^C NMR Spectroscopies 

Figure 2 shows ^1^H NMR (DMSO-d6, 500 MHz): δ 0.85 (t, 3H, CH_3_), δ 1.24 (m, 4H, CH_2_), δ 1.45 (q, 2H, CH_2_), δ 2.067 (m, 2H, CH_2_), δ 2.53 (s, 2H, NH_2_), δ 2.71 (m, 2H, CH_2_), δ 3.19 (m, 1H, CHP), δ 5.39 (s, 1H, NH), and δ 6.73–7.16 (m, 10H, Ar–H). 

^31^P NMR peaks (DMSO, 162.0 MHz), the presence of a singlet peak at =19.24 ppm in the ^31^P NMR spectrum (Figure 3a) confirms the existence of the phosphonate moiety associated with the P(O) signal

Additionally, ^13^C NMR (100 MHz, DMSO-d6) show several carbon peaks consistent with each of the different environments of the carbon atoms in the material (Figure 3b). The modification is identified by the disappearance of the (C=O) peak for salicylaldehyde identified at around δ193.04 ppm and the appearance of the peak of the (P-CH) chiral carbon atom as a doublet at δ = 57.08 ppm. This doublet is characterized by a large coupling constant (J_pc_ = 149.7 HZ) due to carbon–phosphorous coupling. The aliphatic chain appears at δ (ppm): 14.83 (CH_3_), 21.33 (CH_2_CH_3_), 22.54 (CH_2_), 28.54 (CH_2_CH), 47.34 (CH_2_NH_2_), 48.6 (CH_2_NH), and 57.1(CHP), while the aromatic carbons appear at 114.51, 116.91, 120.92, 123.46, 134.338, 135.49, 150.17, and 151.49 ppm.

#### 2.1.3. Textural Analysis (BET)

Using nitrogen adsorption-desorption isotherms, the BET equation was utilized to identify the specific surface areas, pore volume, and pore diameter for the synthesized α-AP compound. The isotherms profiles are type II form, according to Langmuir types. In Figure 4 it is clear that the α-AP compound has a surface area of 1.04 m^2^/g, 0.2385 cm^3^/g pore volume, and 68.137 nm pore size.

### 2.2. Weight Loss Measurements

Weight loss measurements were used to determine the corrosion rates of carbon steel samples in both acid solutions that contained different amounts of the α-AP compound. The variation in weight per unit surface area and time can be applied to determine it. When speaking of uniform and universal corrosion processes, this expression is appropriate. The fluctuation of the rate of corrosion in 1 M HCl and 1 M H_2_SO_4_ in the absence and presence of various concentrations of the α-AP compound is shown in Figure 5. It is clear that in every situation examined, Equation (2) was used to determine the protective effectiveness *E*_W_% of the α-AP compound in both acid solutions. The results are shown in Figure 5. It is obvious that the *E*_W_% increase with increasing the concentration till the optimum α-AP compound concentration at 180 ppm (see Figure 6).

### 2.3. Electrochemical Measurements

The inhibition activity of the α-AP compound on the corrosion performance of carbon steel in 1 M HCl and 1 M H_2_SO_4_ was determined using polarization curves (see Figure 7 and Figure 8). The polarization parameters such as corrosion potential (E_corr_) and corrosion current density (*i*_corr_), and the Tafel slopes (βc and βa) beside the corresponding corrosion inhibition efficiency (*E*_P_%) were measured and are listed in Table 1. *E*_P_% was preconceived from the following equation [27]:(1)$$E_{P}\%=\frac{i_{\mathrm{corr}(0)}-i_{\mathrm{corr}}}{i_{\mathrm{corr}(0)}}\times 100$$
where *i*_corr(0)_ and *i*_corr_ are the corrosion rate of carbon steel without and with α-AP, respectively.

### 2.4. Surface Morphological Study and Corrosion Inhibition Mechanism

SEM analysis is a helpful technique for examining the surface morphology of steel following immersion in various media. This approach is especially helpful in describing how a surface-protective organic layer develops to allow for a high level of inhibition resistance. The surface morphology of mild steel surfaces after three hours in two separate solutions, non-inhibited solutions (1 M HCl and 1 M H_2_SO_4_), and those containing 180 ppm of the α-AP compound is shown in Figure 9. 

Additionally, EDX analysis was carried out and examined to determine the elemental composition of the steel samples both before and after the addition of the α-AP compound (see Table 2). 

### 2.5. FT-IR Analysis

In order to further confirm that α-AP protects the steel surface via an adsorption mechanism, the steel surface, after polarization in the corrosive media (1M HCl, H_2_SO_4_) containing 200 ppm inhibitor, was submitted for ATR–FTIR characterization. The acquired spectrum is presented in Figure 10 in comparison with the pure α-AP inhibitor. Pure α-AP shows important peaks due to the broadband seen at 3450 cm^−1^ for α-AP compound corresponding to (NH_2_). The peak at 3046 cm^−1^ for C-H is aromatic. The peak at 2958 cm^−1^ is corresponding to C-H alkane. The peak at 1596 cm^−1^ is related to (C=C-Ar) stretching in the aromatic ring. In addition, the three peaks at 1247 cm^−1^, 1047 cm^−1^, and 752 cm^−1^ are due to P=O, P-O-C, and P-CH, respectively. These peaks are also observed, although with lower intensity, after the sour corrosion in the presence of an inhibitor. The observed peaks eventually confirm the adsorption of the α-AP compound using the N, O, and P heteroatoms and the pi electron in the C=C group of the aromatic ring. This result may explain the detection of more carbon atoms and a nitrogen atom when the EDAX elemental analysis in Table 2 is compared.

### 2.6. Adsorption Isotherm 

Various mathematical correlations for the adsorption isotherms have been proposed by fitting the current data Langmuir equation which is the most fitted one and is given by the following general equation [28]: (2)$$\frac{C_{R}}{\theta }=\frac{1}{K_{\begin{array}{l} \mathrm{ads} \end{array}}}+C_{R}$$
where *K*_ads_ is the equilibrium constant of the adsorption reaction and θ is the degree of metal surface coverage. The amount of metal surface coverage was determined using weight loss measurements using the following relation [29]: (3)$$\theta =\frac{C_{R0}-C_{R}}{C_{R0}}$$

According to Figure 11, the Langmuir isotherm provided the best fit. Straight lines were drawn from the figure, and regression coefficients (R^2^) nearly equal to unity confirm that the data suited to Langmuir adsorption isotherm well. For HCl and H_2_SO_4_ solutions, the values of *K*_ads_ determined from the Langmuir adsorption isotherm are 7.3 and 14.6 × 10^−3^ ppm^−1^, respectively.

Values for the free energy change of adsorption (ΔG_ads_) were calculated from the following expression [30]:ΔG_ads_ = −RT ln (10^6^ K_ads_)(4)
where T is the absolute temperature, R is the universal gas constant (8.314 J K^−1^ mol^−1^), and factor 10^6^ denotes the concentration of water molecules in the solution (in mg/L). For solutions of HCl and H_2_SO_4_, the computed average ΔG_ads_ values are −22.0 and −23.7 kJ mol^−1^, respectively. This suggests that physisorption is the main mechanism for the adsorption of the α-AP compound on steel. The stability and spontaneity of the adsorption processes are indicated by the negative values of ΔG_ads_ for the adsorption of the α-AP compound on steel [31].

### 2.7. Theoretical Calculations

Figure 12 declares the frontier molecular orbital (“HOMO” and “LUMO”) of the α-AP compound. It is clear that for the LUMO orbitals the benzene ring has high electron density while the electron density distributed at the nitrogen atoms and the oxygen as well as the nearest carbons for HOMO. Moreover, E_LUMO_ and E_HOMO_ are important quantum descriptors that are also used to determine the hardness of the inhibitors which show the adsorption ability of the molecule on the metal surfaces. The hardness is calculated ((E_LUMO_ − E_HOMO_)/2). The Mulliken charges in Table 3 illustrate the predicted active centers of the α-AP compound which easily gives its electron to the metal surface.

## 3. Materials and Methods

### 3.1. Chemicals and Materials

Valeraldehyde, ethylenediamine, and triphenylphosphite were provided by Sigma-Aldrich (Saint-Louis, MO, USA). Fluka AG (Buchs, Switzerland) provided acetonitrile and lithium perchlorate. All reagents were used exactly as they were given without any purification.

For the corrosion rate measurements, mild carbon steel sheet samples with chemical composition (in wt. %): 0.243 C, 0.028 P, 0.0036 Si, 0.320 Mn, and the balance Fe were used. The surface area of each sample was about 6 cm^−2^ (for weight loss tests) and 0.354 cm^2^ (for electrochemistry tests). The samples were repeatedly polished with fine-grade emery sheets, disinfected with acetone, cleaned doubly with distilled water, and dried [32].

### 3.2. α-AP Compound Synthesis and Characterization 

Figure 1 described the preparation of the α-AP. The equimolar ratio of valeraldehyde, ethylenediamine, and triphenyl phosphite were dissolved in 10 mL of CH_3_CN. Then and before the addition of 0.02 g of the Lewis acid catalyst (LiClO_4_), the mixture was stirred vigorously for 10 min (at room temperature). The mixture was agitated until the TLC indicated that the reaction was complete. The final product was filtered, cleaned with acetonitrile, and dried by air to produce the corresponding α-aminophosphonate (α-AP). The as-synthesized solid was recrystallized using methanol to purify the materials. Finally, the powder was dried in an oven. 

^1^H NMR and ^13^C-NMR (solvent DMSO-d6) were determined using a JEOL ECA-500II spectrometer. Materials changed proportionally to the associated solvent, expressed in ppm. The ^31^P-NMR spectra were detected at 162 MHz in DMSO-d6 via a BRUKER spectrometer (Japan). The FTIR spectra were recorded within the range of 4000–400 cm^−1^ using a Waltham spectrometer (USA). The pore volume was determined using the BJH method, and the surface area was determined using N_2_-adsorption-desorption isotherms recorded on a Quanta chrome Nova 3200 instrument (Boynton Beach, FL, USA) under a degassing temperature of 60 °C for 3 h. SEM-EDX microanalysis detectors performed a surface morphological test to examine the surface morphology of C-steel (series: 1200 EX II electron microscope: JEOL-JEM).

### 3.3. Weight Loss and Electrochemical Measurements

The carbon steel samples were weighed before being immersed into 30 mL of 1 M HCl and 1 M H_2_SO_4_ solutions with different α-AP concentrations for 3 h. Samples were removed, scrubbed with a bristle brush under running distilled water, and dried. Then, the weight of the samples was recorded again. Three experimental readings were averaged for each reading that is reported using a Mettler H35AR digital analytical balance. The measurements for weight loss were taken at 25 ± 1 °C, and the rate of corrosion (*C*_R_) was calculated using the following relation [33]:*C*_R_ = W/A × t(5)
where W is the average weight loss, A is the total surface area of C-steel samples, and t is immersion duration. 

The inhibition efficiencies (*E*_W_%) of the α-AP in the applied aggressive solutions were calculated using the following equation [34]:(6)$$E_{W}\%=\frac{C_{R0}-C_{R}}{C_{R0}}\times 100$$
where *C*_R0_ and *C*_R_ are the corrosion rate of carbon steel without and with α-AP, respectively.

A typical glass cell with three electrodes was used for the electrochemical procedures. The counter electrode was a huge platinum sheet, and the reference electrode was a saturated calomel electrode (SCE) [28].

### 3.4. Theoretical Calculations

Different quantum descriptors for the α-aminophosphonate were obtained using density functional theory (DFT) with B3LYB and 6-311G basis set after geometry optimization (see Figure 2). The energy of both the highest and lowest molecular orbitals, EHOMO and ELUMO, respectively, were evaluated. Then, the energy gap Eg was calculated (Eg = ELUMO-EHOMO) [35]. 

## 4. Discussion

The starting materials of the three components in Figure 1 show the synthesis reaction. The detailed solidification mechanism is explained as follows: Herein, the reaction was carried out in a polar aprotic solvent such as acetonitrile (CH_3_CN) with reactants bearing polar groups and non-polar ones; all dissolved well to form a homogenous solution. Once the product started to form (after the addition of the catalyst) the hydrophilicity decreased; indeed, the synthesized products became progressively hydrophobic and began to precipitate from the solvent phase. The different analysis mentioned in the result part confirms the synthesis of the α-aminophosphonate molecule.

From weight loss results obtained in the present study, the corrosion rates drop as the concentration of the α-AP compound rises. This declares that the α-AP compound slows down the rate at which carbon steel corrodes in both acids. The concentration of the α-AP compound chemical and the characteristics of the corrosive media affect how much inhibition occurs.

Electrochemical measurements show that the α-AP compound plays a significant role in the suppression of corrosion of carbon steel in both HCl and H_2_SO_4_ acids, per the data in Figure 7 and Figure 8 and Table 1. This was unmistakably inferred from the fact that the addition of α-AP compound decreased the corrosion current density i_corr_. It is worth noting that the maximum *E*_P_% (s% in the case of HCl and s% in the case of H_2_SO_4_) was obtained at 180 ppm. Both anodic and cathodic Tafel slopes (βa and βc) change upon adding α-AP compound to the medium. This shows that cathodic and anodic reactions both take longer when their processes change [36].

The corrosion inhibition mechanism can thus be clarified using a physisorption system and the nature of the connection between the α-AP compound and the steel surface. Considering that the α-AP compound slows the corrosion process primarily by increasing the extent of steel surface coverage by α-AP compound molecules.

Since steel is positively charged in both uninhibited and inhibited acid solutions, the molecules of the α-AP compound can adsorb at the metal-solution interface [37]. The primary factor affecting the rate at which steel specimens corrode in acidic electrolytes is the tendency of the α-AP compound to adsorb on the surface of steel through their heteroatoms (O, N, and P) [38]. Additionally, the lengthy alkyl chain of the α-AP compound enhances its surface coverage, providing increased surface protection and isolation from corrosive solutions. Studies on surface morphology and electronic configuration have supported this suggestion.

As Figure 9 shows, the SEM micrographs differ significantly from one another. From Figure 9a,c, without an inhibitor we can observe that the steel surface is severely corroded and has interior corrosion damage as a result of the acid solution’s quick corrosion attack; hence, the steel surface becomes rough and inhomogeneous. However, Figure 9b,d shows that with the presence of α-AP compound, the surface damage is significantly reduced, indicating surface protection due to the α-AP compound adsorption onto the steel surface.

From EDX analysis the observation is that steel in uninhibited HCl solution (Table 2) contains a large percentage of iron (69.35), oxygen (23.62), and chlorine (1.1) atoms. This suggests that the production of iron chlorides and/or iron oxides was responsible for the corrosion of steel. Comparatively, in the presence of α-AP compound, the percentage of chlorine (0.17) and oxygen (12.42) are significantly reduced, which means a decrease in the density of corrosion active sites. The oxygen percentage which appears in the presence of α-AP compound is due to the fact that α-AP compound contains oxygen atoms and it coordinated with Fe. Additionally, a new peak of nitrogen and phosphorus was observed with the addition of the α-AP compound, confirming the presence of the inhibitor under study on the steel surface. This inhibitor inhibits iron corrosion by adsorbing to the surface, blocking surface damage, and preventing the formation of ferric and ferrous chloride compounds.

Additionally, a significant portion of the iron (57), oxygen (30.13), and sulfur (2.34) atoms in the uncontrolled H_2_SO_4_ solution are found. Comparing the spectrum of this proportion to that of the sample that contains α-AP compound, it can be seen that the peaks of oxygen (18.27) and sulfur (1.35) are greatly diminished. At the same time, a new one related to nitrogen and phosphorus also appears. All of these findings suggest that the addition of the α-AP compound reduces the corrosion of steel. Given the presence of aromatic rings, phosphorus, and amino groups, which enable the adsorption of the tested molecule to the surface and hence ensure excellent resistance against corrosion; these results are not surprising. The results of the surface characterization analysis support the outcomes of the electrochemical and weight loss tests.

It is well known that frontier orbitals in the adsorption process should be considered to forecast the adsorption site of the molecules under study. In addition, their energy values should consider that the greater the E_HOMO_ (the energy of the highest occupied molecular orbital) is, the more remarkable a molecule’s propensity to donate electrons to the vacant 3d orbital of the metal; however, the molecule’s potential to take e’s from metal decreases as the E_LUMO_ energy of the lowest unoccupied molecular orbitals decreases. The lower value of both molecular orbitals (E_HOMO_ = −9.225 eV and E_LUMO_ = −1.272 eV) indicates that α-AP compound has the ability to donate its e’s to the vacant d orbital of the carbon steel and accept e’s resulting in strong adsorption of α-AP compound onto the surface. Additionally, the lower value of the energy gap (E_g_ = 8.153 ev) confirmed the easier adsorption of the investigated inhibitor to the metal surface [39]. The lower value of the hardness (−3.976 eV) declares that the α-AP compound is easily adsorbed onto the surface of the steel. Concerning Table 3, it is clear that oxygen, nitrogen, and the nearest carbon atoms are the most negatively charged atoms indicating their higher ability for donation by attacking the metal surface. Comparatively, HOMO (see Figure 12) was primarily concentrated in the region around the oxygen, nitrogen, and carbon atoms, which confirms the previous suggestion where the majority of the bonding takes place.

## 5. Conclusions

In the current research, a novel corrosion inhibitor for carbon steel for both hydrochloric and sulfuric acid solutions was established using α-aminophosphonate (α-AP). The α-AP was successfully prepared using a simple procedure and distinguished by FT-IR, ^1^H NMR, ^31^P NMR, and ^13^C NMR analysis. According to the research, the morphology of the α-AP compound is cubic, and its particles have a size range of 17 to 23 μm. It has been found that the substance α-AP compound markedly inhibits carbon steel corrosion for both HCl and H_2_SO_4_ acids. This was clearly implied by the observation that the rate of corrosion significantly reduced when α-AP compound was added. The greatest inhibition efficiency (92.4% for HCl and 95.7% for H_2_SO_4_) was generated at 180 ppm. According to SEM and EDX, α-AP inhibitory layers were adsorbed on the surface of the carbon steel, which gave it remarkable anticorrosion performance. The Langmuir adsorption isotherm is used to categorize the adsorption of the α-AP compound in great detail. The tendency of the α-AP compound to adsorb on the surface of steel through their hetero-atoms (O, N, and P) is the main factor influencing the rate at which steel specimens corrode in acidic electrolytes. A modern research pathway for creating cutting-edge ecologic corrosion inhibition processes may be made possible by the newly constructed α-AP compound, which is a cheap and efficient corrosion barrier.

## Data Availability

Not applicable.

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
