# Peer review of "Novel Corrosion Inhibitor for Carbon Steel in Acidic Solutions Based on α-Aminophosphonate (Chemical, Electrochemical, and Quantum Studies)"

_molecules, 2023, doi:10.3390/molecules28134962_

Round 1

Reviewer 1 Report

The authors prepared α-aminophosphonate as a corrosion inhibitor for carbon steel in two different acid solutions.  Besides, physiochemical and electrochemical characterization were also conducted to investigate the anticorrosive mechanism. A computational chemical approach was used to support the experimental results. Although the subject is interesting and the synthesized inhibitors showed a good inhibition effect, there are some problems with this paper that should be addressed before it is considered for publication. They are outlined below:

1.      The first lines of the abstract (20-23) are not necessary. The abstract must be concise.

2.      What type of carbon steel did you use?

3.      Table 1. You stated that "Three experimental readings are averaged for each reading". Do you have the standard deviation?

4.      In Figure 10, it's not possible to read the scale. Also, in the eds analysis, figure 11, the letters are blurred

5.      Why didn't you run any EIS tests?

6.      The temperature was not taken into account. No surface morphology results, except SEM-EDS, were presented. Product analysis should be conducted including XRD, and/or XPS, etc.

These results lack detailed mechanism explanations. Hence, the discussion and conclusion must be improved also. The results indicate that the adsorption mechanism is a mixed-type adsorption (i.e., physical and chemical adsorption), however, it seems that physical adsorption is much more dominant than chemical one. Yet, you didn’t describe this mechanism. From your discussion, (lines 351-358) it seems that the absorption mechanism is due only to the chemical one.

Author Response

Response to Reviewer 1:

  1. The first lines of the abstract (20-23) are not necessary. The abstract must be concise.

Reply: According to your point of view, the lines (20-30) have been removed from abstract in order to be concise.

  1. What type of carbon steel did you use?

Reply:  type of carbon steel is mild steel and this mentioned in the experimental part (line 91)

  1. Table 1. You stated that "Three experimental readings are averaged for each reading". Do you have the standard deviation?

Reply:  according to comment of one reviewer, the Table 1 was omitted. The data and standard deviation are presented in Figs. 5 and 6 in revised manuscript

  1. In Figure 10, it's not possible to read the scale. Also, in the eds analysis, figure 11, the letters are blurred

Reply:  We are so sorry about that, the images in figure 10 (figure 9 in revised manuscript) were replaced with higher quality ones, so it became clearer to read the scale. According to comment of one reviewer, the Fig. 11 was omitted It is also described in detail in table 2. All figures are very clear in supplementary information file

  1. Why didn't you run any EIS tests?

Reply:  we used two different methods to determine the inhibitors efficiency i.e., chemical and electrochemical methods. The two methods are comparable and confirm our results. We fully agree with the reviewer comment that the EIS is the one from electrochemical methods is very useful. But in this work, we used polarization method to confirm the weight loss method. Surly, we will take in our consideration the EIS method in the future work

  1. The temperature was not taken into account. No surface morphology results, except SEM-EDS, were presented. Product analysis should be conducted including XRD, and/or XPS, etc.

Reply:  all the experimental works were conducted at 298 K. surface morphology results were investigated by using SEM-EDS.  We done FTIR for the steel surface, after polarization in the corrosive media (1M HCl, H2SO4) containing 200 ppm inhibitor This part is explained in detail in result part (3.5. FT-IR analysis) (line242)

See supplementary information file.

  1. These results lack detailed mechanism explanations. Hence, the discussion and conclusion must be improved also. The results indicate that the adsorption mechanism is a mixed-type adsorption (i.e., physical and chemical adsorption), however, it seems that physical adsorption is much more dominant than chemical one. Yet, you didn’t describe this mechanism. From your discussion, (lines 351-358) it seems that the absorption mechanism is due only to the chemical one.

Reply:  we correct the adsorption mechanism which is mainly physical adsorption. The discussion and conclusion were improved and highlighted in manuscript (lines 292,293,340-351)

Reviewer 2 Report

Inhibitors made of organo-phosphorus compounds and their derivatives are an effi- cient way to reduce the risk of corrosion in oil fields without posing any environmental risks. There are results of α-AP like CR and EW and other data, which is helpful for us to get more information on the corrosion inhibition of pipe lines. After answering the following questions, it can be published in moleculars:

 1) Scheme 2. The optimized structure of α-AP, which should give the atom element names.

2)  Fig. 1, there is no sub-figure labels. And the fonts in the SEM images are too small.

3)  Figs. 3 and 4, the captions are not completed. And the curves are necessary to be redrawn. Here, the attitude to the exprimental data is not so strict.

4) Fig. 5 is too unclear, which seems copied from other places rather than authors' own measured data.

5) Figs 6 and 7, the curves are deduced from eqs. (1) and (2). and CR seems a simple relationship with EW, but the comparison of the first half curves of H2SO4 and HCl solutions is different from the situation of the second half curves. Why?

6)  Table 1 can be omitted, since the content appears in the above diagrams.

7) Table 2 can be transferred into curves, which would give a more readable picture.

8) Figure 11 can be rewritten in an table. The present shape is unclear and ugly.

9) Fig. 13, HOMO and LUMO should be briefly explained, which is helpful for the readers in the exprimental field.

10) The journal titles in references should be in a unified format.

ok

Author Response

Response to Reviewer 2:

 1) Scheme 2. The optimized structure of α-AP, which should give the atom element names.

Reply:  the atom element names were added in the optimized structure of α-AP (Scheme 2)

2)  Fig. 1, there is no sub-figure labels. And the fonts in the SEM images are too small.

Reply:  sub-figure labels were added and The SEM image was improved (see Fig.1)

All figures are very clear in supplementary information

3)  Figs. 3 and 4, the captions are not completed. And the curves are necessary to be redrawn. Here, the attitude to the experimental data is not so strict.

Reply:  The quality of the Figs 3 and 4 (figure 9 in revised manuscript) were improved.  

4) Fig. 5 is too unclear, which seems copied from other places rather than authors' own measured data.

Reply:  the Fig. 5(figure 4 in revised manuscript) was re-drawing from the data. It is now clear  

5) Figs 6 and 7, the curves are deduced from eqs. (1) and (2). and CR seems a simple relationship with EW, but the comparison of the first half curves of H2SO4 and HCl solutions is different from the situation of the second half curves. Why?

Reply:   this different come from the different in the corrosion rate of blank solutions (1 in 1M HCl and 1M H2SO4) which leads to the different from

6)  Table 1 can be omitted, since the content appears in the above diagrams.

Reply:   In compliance with your opinion, Table 1 has been removed from the text as it appears in the above diagrams.

7) Table 2 can be transferred into curves, which would give a more readable picture.

Reply:   This table contains many of polarization parameters for carbon steel in 1.0 M HCl and 1.0 M H2SO4. So, it is very difficult to transfer into curves. This needs many of figures. We need to keep it in one table 

8) Figure 11 can be rewritten in an table. The present shape is unclear and ugly.

Reply:   Figure 11 is already written in Table 2. The Figure 11 was removed 

9) Fig. 13, HOMO and LUMO should be briefly explained, which is helpful for the readers in the experimental field.

Reply:   HOMO and LUMO was explained in the experimental field (lines 301-306) also explained in results and discussion sections and highlighted in manuscript (lines 381-397)

10) The journal titles in references should be in a unified format.

Reply:   All references have been reviewed to be consistent with the journal's tablet. It was confirmed that all journal titles are in a unified format, and it has been modified in the manuscript.

Round 2

Reviewer 1 Report

The amendments made by the authors in response to the reviewers' comments are adequate, hence the paper is accepted for publication.

Reviewer 2 Report

my comments are addressed successfully.